# Time and Change in Advaita—Gauḍapāda in Dialogue with Vasiṣṭha and Nāgārjuna

Sthaneshwar Timalsina

Asian and Asian American Studies, Stony Brook University, Stony Brook, NY 11794, USA;
sthaneshwar.timalsina@stonybrook.edu

**Abstract:** In the classical philosophical landscape of India, the Advaita of Śaṅkara occupies central stage. Besides the Upaniṣadic literature, the *Gauḍapāda-kārikā* (GK) of Gauḍapāda is the primary text in this school. Relying primarily on the GK, this essay explores the ways the issue of change can be addressed within the Advaita paradigm. For Advaitins, there exists only the singular reality of Brahman, of the character of non-differentiated consciousness. In this paradigm, the attributes of both being and blissfulness never change. Furthermore, the central teaching of Gauḍapāda is the doctrine of 'non-origination' (*ajāti*), that nothing is ever originated. For Advaita, change or deviation is possible only under the spell of illusion, as the absolute is changeless. By comparing the position of Gauḍapāda with other classical, non-dual philosophies, this paper explores arguments for and against change in the classical philosophical school of Advaita.

**Keywords:** Gauḍapāda; Advaita; change; svabhāva; Śaṅkara; Yogavāsiṣṭha; Nāgārjuna





## 1. Introduction

From the perspective of Advaita, all that exists is Brahman, the absolute. Upon the question concerning phenomenal experience, Advaitins adopt either a two-tier analysis of reality, similar to what Nāgārjuna does, and say that the reality of everyday experience does not bear the ultimate truth, or adopt a triadic structure, incorporating the conventional, making everyday reality not quite fictional but still not bearing the ultimate truth. In order for us to engage the category of change in the Advaita platform, we therefore have two different models, as the Advaitins address this issue either by adopting the three-tiered structure that change is integral to the conventional, or by adopting a two-tiered structure with change corresponding to the fictional. The actual difference is that to say something is conventionally changing is not the same as arguing that change is mere illusion. By focusing on Gauḍapāda for addressing the issue of change and by bringing into this dialogue the Buddhist philosophers Nāgārjuna and Vasiṣṭha, the mythical narrators of the voluminous philosophical epic, the *Yogavāsiṣṭha* (YV), I have chosen the trajectory of two-tiered analysis of reality[1]. The objective here is not to argue that these three philosophers have the same arguments to make. Quite the contrary. By grounding the notion of change in Gauḍapāda's verses, the *Māṇḍūkya-Upaniṣad-Kārikā* (MUK), which is otherwise known as *Gauḍapāda-kārikā* (GK), I shall explore the parameters to determine the extent to which MUK overlaps the Nāgārjunian thesis with regard to the issues of time and change. What I envision here is a trialogue: a conversation among Nāgārjuna, Gauḍapāda, and Vasiṣṭha. All in all, this is an analysis of change from what the Advaitins call "the ultimate viewpoint" that makes no categorical differentiation between the conventional and fictional. To adopt this line of argument is to underscore the thesis that time and change in waking consciousness are akin to dream consciousness, or that the world of convention and the entities experienced during dreams or hallucinations do not bear a different degree of reality.

The topic of this conversation is change. From the perspective of two-tiered reality, change in our commonsense reality is akin to change in dream or fantasy. Both MUK

and the YV repeatedly borrow examples of dreams and hallucinations to make this point. Some scholars have stretched too far on this basis and argued that the philosophy of YV is identical to that of Gauḍapāda[2]. However, if we closely analyze the narratives in the YV, we come to realize that the philosophy addressed here is more nuanced and cannot therefore be reduced to Gauḍapāda's philosophy. In fact, YV is an assemblage of multiple streams of contemporaneous thoughts, be that Buddhist Mādhyamika or Yogācāra streams of thought, or the Advaita of Śaṅkara, or Trika philosophy as espoused by Utpala and Abhinavagupta. Nevertheless, it would be wrong to assume that the text is a haphazard mixture of different and occasionally contradictory thoughts, since it weaves different philosophies within the overarching thesis of singular self-given consciousness while using narratives as illustrations, making it possible for different models of non-dualism to converge. In this conversation, I shall employ a few selected narratives from the YV to evaluate some nuances of Gauḍapādian philosophy. This will help us configure the ways in which the category of change can be addressed in the nondual platform. And when we compare, we will notice not only the similarities, but also some conceptual gaps.

## 2. Gauḍapāda on Time and Change

Gauḍapāda is widely known for two reasons: first, for his stanzas upon the *Māṇḍūkya Upaniṣad* (MU), and second, as the grand master of Śaṅkarācārya (eighth century). Philosophically speaking, Gauḍapāda is known for the doctrine of 'non-origination' (*ajātivāda*), that the absolute never deviates from its original form. It is contradictory to argue for change while relying on this platform. We can nonetheless address change by bringing to light the conceptual gap between MU and MUK. Even though MU is a very concise text, composed merely of 12 stanzas, there are sufficient hints to contrast the central premise of Gauḍapāda.

The *Māṇḍūkya Upaniṣad* (MU) maps all that exists within the single phoneme of OM, segmented into four sections of A-U-M, and the fourth, referring to *bindu* or a nasal sound. In this mapping, the first stanza compares the three sections of OM with three aspects of temporality—the past, present, and the future (stanza 1). The same stanza also confirms that whatsoever transcends the triadic structure of temporality is also the very phoneme of OM. When the MU outlines three different states of subject, with the subject of the waking state enjoying the externals (stanza 3), the subject of the dream state consuming the differentiated objects (stanza 4), and that of the deep sleep merely enjoying bliss (stanza 5), these subjects are not considered illusory but as the very modes of the self that also transcend this triadic structure. If we read closely, the text appears to be supporting a philosophy that confirms the dyadic structure of the absolute, having both the manifest and unmanifest forms. If the fourth or the formless is the transcendent state, the manifest modes in the triadic structure are its immanent modes. Otherwise, it would be contradictory to address the transcendent state as the motivator of all (*sarveśvara*, stanza 6) and the knower of all (*sarvajña* stanza 6). Furthermore, this absolute is addressed here as the substratum for all of the entities to come into being and to return back (*prabhavāpyayau hi bhūtānām*, stanza 6).

We will notice a clear contrast if we compare the third and the fourth chapters of MUK. These chapters do not confirm the fourfold structure of the self or analyze the absolute in transcendent and immanent modes. On the contrary, they simply reject the phenomenal in order to confirm the absolute. These are also the chapters where Gauḍapāda's originality comes to fullness with their stress on the concept of non-origination (*ajāti*). Gauḍapāda argues here that "that what [appears as] being originated, does not come into being" (ch. 3, verse 2). The central premise of this paper comes from Gauḍapāda's proclamation that nothing that appears to be manifesting does in fact manifest (… *na jāyate kiñcij jāyamānaṃ samantataḥ*, ch. 3, verse 2.), meaning, change is a mere mental projection and does not reflect the reality as it is. When this paradigm is engaged, all that exists is motionless and changeless Brahman or pure consciousness, with all the manifest structures or subjects and their corresponding objects being merely a projection of ignorance. The central metaphor that Gauḍapāda introduces in this chapter is that of the sky (ch. 3, verses 3–9), which

identifies the absolute or the self as the sky, with an underlying assumption that the sky cannot be fragmented, differentiated, transformed, or polluted, and that it is birthless and deathless.

In order for us to ground temporality, we need to establish change, and for that we need to establish difference. In the absence of difference as a category, there is no means of establishing temporality. Cause and effect are different. Now and the past or now and the future are two different modes of temporality. The cause precedes its effect. Only on these grounds can we conceive of temporality. If there is no chain of causality, if there is nothing prior and nothing posterior, if there is no difference in any accord, there is no temporality. In order to address the ways Gauḍapāda rejects change, it is necessary that we address the very notion of *ajāti* or non-origination in his philosophy. Gauḍapāda argues:

> This [absolute] is distinguished merely due to *māyā*. Otherwise this unborn cannot [be differentiated] in any account. If it were to be differentiated in reality, the birthless would succumb to extinction.

> Those adhering to dogmas (*vādins*) anticipate origins of the essence that is birthless. How can the essence that is birthless and deathless attain mortality?

> Neither a mortal can be immortal, nor an immortal, a mortal. It is never possible for an intrinsic nature to be otherwise. (GK III.19–21).

> Through these statements, Gauḍapāda makes the following arguments:

(i)   There is no substantiality of the categories such as cause and effect or birth and death, as these do not reflect the nature of the reality.
(ii)  The absolute does not lack its own intrinsic nature or *prakṛti*, in other words, *svabhāva*.

Even while contradicting change or causality and thereby rejecting temporality, Gauḍapāda is not rejecting *svabhāva* or *prakṛti*. When rejecting change, Gauḍapāda is confirming something being changeless. For otherwise, his absolutism would be identical to that of Nāgārjuna.

To conceive of temporality against this background is to acknowledge the premise that time measures change or that time marks difference. In other words, difference is integral to change. Accordingly, something cannot change and not be different from its primordial form. If an entity that is constant were to change, it would then be transitory, changing from its original form. Meaning, the entity was not constant to begin with. In order for us to conceive of time, we therefore need a paradigm of metaphysical difference, something that is not conceivable in the absolute of Gauḍapāda[3]. His rejection of temporality therefore can be considered only provisional, rejecting what others believe to be the case, confronting their own presuppositions. But we cannot interpret this to be the thesis of Gauḍapāda himself. Negation of temporality in this paradigm is in the absolute sense. Gauḍapāda is not arguing that there are occasions where temporality is not applicable. Time and change, according to Gauḍapāda, are merely figments of the imagination, superimposed due to *avidyā*. The rejection of origination is therefore also the rejection of temporality.

## 3. Rejection of Origination

By adopting the dialectical methods of Nāgārjuna, Gauḍapāda rejects origination both on the grounds that "something that is" (*bhūta*) cannot come into being, and likewise, "something that is not" (*abhūta*) cannot come into being either (GK IV.3–4). The argument is that it does not make sense for something that already exists to come into being, and something that does not exist, like the horns of a rabbit, cannot be brought into existence. This twofold rejection makes perfect sense in Gauḍapāda's paradigm, for he accepts the category *svabhāva*, that whatever exists has its intrinsic nature and that what exists does not exceed its nature. Even though many of these arguments in rejection of change or origination can be traced back to Nāgārjuna, his is not a paradigm that accepts the absolute with its own intrinsic nature (*svabhāva*). On the contrary, Nāgārjuna is seeking to ground the thesis that the category, inherent nature (*svabhāva*), does not exist. When Nāgārjuna rejects change or motion, he assumes that for an entity to exist, it would need to have its

own intrinsic nature (*svabhāva*) and since change or motion is possible contingent upon the factors that change or that move, it therefore lacks its own intrinsic nature. However, it would be wrong to assume that Nāgārjuna is proposing that only entities that are endowed with an intrinsic nature can exist. He is only rejecting the thesis that motion or change exists. Since he is not proposing something, he does not need to establish it. In his account, those who are proposing a thesis that there are entities who have their own intrinsic nature are the ones who need to establish it[4]. At this juncture, Gauḍapāda departs from Nāgārjuna. As far as Gauḍapāda is concerned, there exists the absolute, pure consciousness, or the Brahman, having its own intrinsic nature. And when Gauḍapāda rejects change, he has this absolute nature in mind that he asserts as changeless. For both Nāgārjuna and Gauḍapāda, there is no change. However, Nāgārjuna believes that change as a category cannot be established, whereas for Gauḍapāda, the absolute is changeless, or that changelessness is the *svabhāva* or the nature of what is, and for it to change would imply that what is as it is has lost its intrinsic nature. Gauḍapāda therefore can make a propositional claim of his negation of change, and likewise, he can make a propositional claim of the singularity and consistency of the absolute. On the other hand, Nāgārjuna cannot make a proposition as such. And this is why he confesses that "if I had any proposition, it could have been rejected"[5]. As far as Gauḍapāda is concerned, he uses the terminology of both *prakṛti* and *svabhāva* (e.g., GK IV.7–8) when declaring that one cannot alter the intrinsic nature of what constitutes an entity as it is. Basically, if $H_2O$ is the intrinsic property of water, we cannot conceive of water and not have $H_2O$. Gauḍapāda defines *prakṛti* in the following terms:

> *Prakṛti* is such that it does not relinquish its inherent nature. This *prakṛti* is self-given (*sāṃsiddhikī*), is the being of and for itself (*svabhāva*), it is innate (*sahaja*), and it is not constructed (*akṛta*). (GK IV.9).

While the scholars comparing the philosophy of Gauḍapāda with that of Nāgārjuna and other Mahāyāna Buddhists[6] have brought to light the overlapping philosophical terrain, they have broadly ignored the factors that cannot be reconciled. For Gauḍapāda, there is an absolute, the Brahman, and as for its intrinsic characteristic, it is a self-aware pure consciousness devoid of any differentiation. If Gauḍapāda is adopting the dialectical method, this does not imply that he is endorsing Nāgārjunian metaphysics. For Nāgārjuna, rejecting *svabhāva* is at the core of his argumentation, as his philosophy emerges in negation of the Sarvāstivāda claims regarding *svabhāva*. Gauḍapāda, accordingly, is confirming the positive being of the changeless absolute.

The central argument of Gauḍapāda is the thesis of non-origination (*ajāti*). The rejection of temporality in this account therefore rests on the very premise of causality. Gauḍapāda anticipates from his opponents who adhere to temporality that time is intrinsic to the very notion of causality, and when one rejects causality, one is also simultaneously rejecting temporality. Gauḍapāda rejects causality by demonstrating circularity in the argument, that for something to be conceived of as the cause, it anticipates an effect, and for something to be an effect, it anticipates its cause (GK IV.14, 18). In absence of the givenness of an effect, one cannot confirm something to be the cause, as the very effect depends on the givenness of the cause. Cause and effect, therefore, are superimposed on what it is. And if this is the case, the notion of temporality cannot be confirmed, since causality is not confirmed. Time in this paradigm is basically an order; it confirms the sequence of something being prior in relation to the next being posterior. The notion of causality makes sense as long as we recognize that the cause precedes the effect. Even for the notion of simultaneity, we need order. In absence of succession, the concept of simultaneity makes no sense. Meaning, if we did not have the concepts of prior and posterior, we would also lack the concept of simultaneity, and these are the only signs to confirm the modes of temporality. Gauḍapāda's rejection of change stands on this very premise, that the order of succession between cause and effect cannot be confirmed; in absence of order, the notion of simultaneity ceases to make sense, and as a consequence, the concept of time will lack its reference. We glean this all from Gauḍapāda's statement that "you have to seek an order of succession only upon the possibility of cause and effect" (GK IV.16). What Gauḍapāda

is saying is that time is not self-given and that the concept of temporality is introduced merely to explain causality. However, in reality, causality cannot be explained. Accordingly, anything that rests on causality crumbles on its own accord. When an entity is cognized as being originated, one can argue that this "being originated" as a property is intrinsically given alongside the manifestation of an entity. However, if the entity were to be given to consciousness upon being originated, consciousness should also grasp its cause at the same time. However, the cause is not self-given by the mere conceptualization that an entity is originated (GK IV.21). Śaṅkara expands upon Gauḍapāda's lines in following words:

> If an entity is cognized as coming into being, how is it that the cause that antecedes it not cognized? One who cognizes an entity as coming into being should also cognize its cause[7].

The point is, when we claim that all composites are originated, we are taking for granted that the composite entities have some cause. However, even when we do not know what caused it, we can know that something is caused. When Gauḍapāda rejects causality on the basis that the cause is unknown, he is also rejecting inferential knowledge and resting his argument on direct apprehension.

Gauḍapāda adopts Sarvāstivāda vocabulary when he initiates the conversation on temporality. For instance, the category *adhvan* or the three courses of time (*adhvan* GK IV.27) is more common to the Sarvāstivāda. Temporality referred to here is the "course" of an entity coming into being, enduring, and ceasing to exist. Accordingly, the three modes of time are nothing but the very modes of an entity. In other words, time as such does not exist and it is only a device to explain the presentation of an entity in different modes. This becomes further elucidated when Gauḍapāda explains that something that is not there in the beginning (*ādi*), or in the end (*anta*), does not exist in the present mode of time (*vartamāna*) either (GK IV.31). This verse is also a rejection of the Sautrāntika position that entities exist only for the present instant and are inferentially given. The doctrine of *māyā* helps to explain the contradiction in Gauḍapāda's paradigm that entities of the waking reality are not substantially present in the way they are given but appear as such only due to *māyā*. In this way, *māyā* resolves the contradiction that entities may appear and not exist at the same time.

One additional issue on temporality is that the present moment is given in the sense of endurance and is not as ephemeral. We experience endurance, we experience entities presenting themselves, and we impose cessation based on the memory of endurance by making sense of the temporal flow. If time is a category for explaining our experience, it is meaningful only when we accept its three-partite structure. Otherwise, the mere being of the present mode, or, for that matter, any other mode, does not help us ground our everyday experience. When Gauḍapāda reduces external entities into mental projections, he is collapsing time within mental entities, rejecting the argument that time has its own extension. Gauḍapāda uses duration in dream time to reject temporal extension. He argues that just like there is no real extension in temporality in a dream, the same should also be applied to waking consciousness. When Gauḍapāda argues that there is no extension of time in dreaming (*adīrghatvāc ca kālasya* GK II.2), he is rejecting the convention that time has actual duration. By conflating reality as grasped in dreaming with that of waking consciousness, he successfully makes the argument that there is no categorical difference between entities that are experienced in dreaming and those given to waking consciousness. Gauḍapāda further argues that, when it comes to dreaming, "there is no temporal law" (*kālasyāniyamād* GK IV.34). Now, we cannot read the category of change outside of the parameters that Gauḍapāda provides for temporality. Any change that is outside of temporality is non-change, or non-deviation, which Gauḍapāda includes within the category of *Ajāti* or non-origination.

## 4. Gauḍapāda on Change

Since no actual origin and no actual temporality can be conceived of within Gauḍapāda's non-duality (*advaya*), there is no real change either. In essence, change can be conceptu-



alized only within the mental horizon. With this underlying premise, Gauḍapāda says, "the almighty [self] (*prabhu*) constructs by means of manifesting itself in varied forms the entities that are situated within mind and by making the objects outside of the mind as having a fixed [spatio-temporal] law (*niyata*)" (GK II.13). The etymology Śaṅkara gives to explain "*vikaroti*" as "*nānā karoti*" or "diversifies" underscores the power intrinsic to the self to project mental predispositions into outside world in the form of variegated objects. There is no compelling reason to read that the process outlined here—seminal entities residing in dreams and being projected outside in variegated forms—is necessarily a confirmation of illusionism, for it can be interpreted both ways: that dream objects are unreal and therefore anything that has dream as its foundation is unreal, or that dreams are an expression of potentials intrinsic to the self and are an expression of the magical force of *māyā*, and that the externals are the blossoming states of what lies within the mind in the seminal form. So far, we have read Gauḍapāda only along the lines of illusionism, that Gauḍapāda's identification of external reality with dream objects is meant to reject essentiality of the externals. However, we can counter this paradigm and argue that a dream is a magical projection of the intrinsic potential of the self and the external world is its materialization. When we continue reading the second chapter from GK, the very next verses assign external or internal entities as '*kalpita*', that is, imagined (GK II.14–16). But again, we are reading imagination as sheer fantasy bearing no reality. However, the word "*kalpanā*" or derivatives of the root $\sqrt{k\!l\!rp}$ can mean both imagined or constructed. GK II.17 explains this process with the term of *vikalpita*, where the etymology is that of differentiation or diversification. This is not about imagining something that is not there but about imagining the manifold where in reality there lies a singular reality. Again, to say that the singular reality manifests into the manifold is not identical to the statement that all manifestations are mere illusions.

Gauḍapāda Proposes Two Different Modalities to Explain Change:

These are the illustrations to explain origination. Just like the sky [imposed to be different] from the sky [reflected] within pitchers, the self is likewise [differentiated] with the individual selves. Or, it is just like the pitcher [differentiated due to it] being a composite[8].

The first illustration rejects any actual change in the modality of the base entity. The sky does not change into anything but still gives the appearance of the manifold, depending on different substrates. In the second illustration, pitchers are actually nothing but clay, but even then their modality changes, and so depending on different presentations of the same base material, we identify some objects as pitchers and others as vases, etc. Even though Gauḍapāda is not trying to explain origination, the two models he provides explain two different ways for Advaita to explain the manifold. In the first one, differentiation is merely a projection of the mind and there lies no difference as such. In the second case, difference in the modes of presentation of what it is does not constitute difference in the absolute. In the first case, the difference imposed on the sky collapses alongside the destruction of the pots. In the second case, even while the pots endure, one can recognize non-difference in their causal form. Now, reflecting upon change, we have come up with two different interpretations, that either change is a mere projection or that all the manifest differences that are actualized in the manifold rest on the singularity of the absolute.

The Gauḍapādiyan concept of *advaya* helps us explain what he means by non-origination (*ajāti*). In every cognitive mode, we have the dyad of subject and object (*grāhaka* and *grāhya*), and this bifurcation of consciousness is what makes it possible for us to cognize an object, the part of consciousness that assumes the pole of objectivity, whereas the other assumes subjecthood. Gauḍapāda explicitly says that both these poles, whether in waking or dreaming states, are nothing but the pulsation of the very mind:

Just like the appearance of the dyad in the dream [in terms of subject and object] is nothing but the very pulsation of the mind due to *māyā*, the appearance of the dyad in the waking state is also likewise the pulsation of the mind due to *māyā*[9].

In essence, his model of Advaita is based on rejection of the dyadic structure of subject and object, that consciousness as it is does not split in terms of these polarities,

and it is only due to *māyā* that such a projection occurs. We need to keep in mind before proceeding further that the stanzas like the ones above that compare objects in the waking state to those in dreams are meant to reject the phenomenality of the externals. However, we can read the same stanza as affirming what is given, reducing both the object and subject consciousness to consciousness itself. And this makes a big difference when we distinguish Gauḍapāda's philosophy from that of Nāgārjuna, as it opens the space for us to contextualize the philosophical parameters for the narratives in the *Yogavāsiṣṭha*. Conceptual ambivalence is paramount in the YV paradigm of magical realism. The same dream analogy can be read here as rejecting the substantiality of the waking or confirming the substantiality of the dream entities. Unlike Gauḍapāda, YV stresses the creativity of consciousness, and therefore, if waking is one extension of consciousness, so also is dreaming. And again, both do not contradict the foundational nature of consciousness.

### 5. Three *Yogavāsiṣṭha* Narratives on Time and Change[10]

This is not the place for extensively addressing the philosophy in YV. It is not even possible for me to address a single narrative to its fullest extent. This is not even necessary for the current purpose, as we only want to learn how YV addresses the issue of change. It is nonetheless relevant to say a few words so that we can contextualize these narratives in depth. YV subsumes the philosophies of Nāgārjuna and Gauḍapāda, alongside that of Vasubandhu within magical realism, making it possible for something to be real and unreal at the same time, depending on perspective. The philosophy that grounds YV is *dṛṣṭisṛṣṭi*, or that "creation is seeing", again considering "seeing" as a polyvalent term with multiple levels of meaning, referring to what we see, the finite vision of grasping objects, or conceptualization, or in its exalted meaning as the act of consciousness with its expressed capacity to manifest the manifold[11]. In the philosophical paradigm of YV, we need to explore a new meaning for time and space, as the conventional understanding of them as being either real or unreal does not encompass the gravity of time and space as potencies of the absolute that manifest differently corresponding to different strata of subjectivity, giving rise to different strata of objective reality. Like Schrödinger's proverbial cat, the characters in YV can be both dead and alive at the same time. The narratives in the YV are semi-open-ended, meaning, YV itself gives a lengthy philosophical commentary on the narratives and at the same time leaves open the parameters for further analysis. YV grounds the dynamism of being in a unique fashion, wherein the perspectives of singularity versus plurality or change versus permanence stop making sense in their usual applications. There really is no hierarchy of reality here. Just like a turtle is a turtle whether or not its limbs are fully extended out of its shell, water is just the same whether it is flowing through canals or in a reservoir, and singularity or the manifold does not contradict the essential nature of consciousness. Even though YV has many narratives for us to choose, I would like to select just three of them. These are: i. "The story of magic" (*indrajāla upākhyāna*), ii. "The story of Gādhi", and iii. "The story of one hundred Rudras". These narratives epitomize change in space, time, and subjectivity, and these narratives, like any other in the YV, come with their own intrinsic philosophical analyses. When reading the YV, we need to keep in mind that we do not conflate the philosophy within YV in theorizing the narratives. We cannot forget that YV is not just a book of narratives, but a philosophical treatise that uses narratives for illustration.

The first narrative, "The story of magic", is from the section on origination (Utpatti, chapters 104–122; For all the Yogavāsiṣṭha references, Śarmā [1918] 1937). The story centers around King Lavaṇa, who, brought to swoon by a magician, finds himself lost in a jungle. Exhausted and hungry, he takes refuge with a low-caste woman. Married with children, Lavaṇa reconciles his early life with his present circumstances, merging with that of the livelihood of a leather-worker. As time goes by, he finds himself in a deadly famine, incapable of feeding his children. As a gesture of self-sacrifice, he kills himself so that his hungry children can have food. As soon as he jumps into the fire, he wakes up to his early life, only to discover himself at the royal seat, having spent only a few moments. The

storyline takes yet another turn when the king, in his expedition to expand his kingdom, enters a village only to confirm that everything he experienced in the apparent swoon did in fact happen in this village.

The next episode, "The story of Gādhi", is from the section called "Pacification" (Upaśama, chapters 44–49). In this narrative, Gādhi seeks to learn about *māyā* and once, when bathing in river, he falls into a magical swoon and finds himself dead. As the swoon continues, he finds himself reborn as Kaṭañja, a lower-caste person, in a remote kingdom. Some bizarre circumstances lead him to be anointed as the king of this town and he rules there for eight years. The story takes a turn when there is famine and people blame Kaṭañja, as a lower-caste person, for acting sinfully, that is, assuming the position of the upper caste. This charge leads Kaṭañja to kill himself, but as soon as he jumps into the pyre, he wakes up and finds himself bathing in the river. He keeps contemplating upon what he has undergone, as everything felt as real to him as his everyday reality. And the final twist in the narrative comes when a stranger arrives at his place and tells him that he is just travelling from a kingdom where the king killed himself as he was a lower-caste person and the kingdom was suffering from famine.

The first part of these two narratives give a linear reading of phenomenal reality as merely a dream-like projection. But the storyline becomes complex with the twist in the story, when it is reaffirmed to both characters that what they underwent did in fact occur somewhere: Lavaṇa actually discovers the village and Gādhi is assured by a reliable source that his experience corresponded to something real. These narratives epitomize the magical power of *māyā* in parallel realities that both occur and do not occur, depending on the perspective of the viewer.

Let us explore the final narrative before we enter a broader conversation. In "The story of one hundred Rudras", in the section on "Enlightenment" (YV, Nirvāṇa I 62–66), a mendicant fancies himself as a tribesman and his fantasy materializes and he finds himself as Jīvaṭa. This new character dreams of himself as a learned scholar and the scholar in turn as an emperor who fancies himself as a celestial being. In a similar but different chain of subjective transformation, the nymph dreams of herself as a deer, and the deer in turn as a creeper, and the creeper into a bee, and the bee as an elephant. In these multiple, successive chains of transformation, the mendicant eventually dreams of being a swan and the swan finds its identity as Rudra. In this final stroke of dreaming, since Rudra is an omniscient being, this fancied self suddenly realizes the entire chain of manifestations, including his new identity as Rudra. Rudra then enters the realm of creation where the mendicant was asleep and he wakes him up to actualize his real identity. Now, for these two individuals, the mendicant who also is an original dreamer and Rudra, the fancied subject who recognizes his true identity and is awakening the real subject, both enter the realm where the mendicant had dreamt of himself as Jīvaṭa. They find Jīvaṭa asleep like a dead log, dreaming the entire chain of events, and Rudra and the mendicant wake him up. Then, these three subjects move through the realms of the Brahmin and of the king, which all were the fantasy of the mendicant in the first place. In essence, every aspect of the fantasy corresponded to something actual: all of the events and subjects were both real and fantastic at the same time. In this indeterminable field of possibilities, only the paradox makes sense, as everything is both real and fictional at the same time. The issue of change and changelessness, being one and many, being fictional and actual, are all collapsed in this paradigm. All of the characters in this narrative recognize themselves as Rudras, making the group of one hundred Rudras. They are both one and many at the same time. As one, nothing has ever changed, and as many, they actualize their differences with each other.

In all three narratives, the stress is on a foundational state, upon which are layered other dream-like states. In the first two narratives, the subjects carry on their phenomenal personalities, only to awaken from their projected subjectivities, whereas in the last one, the projected Rudra awakens other subjects from their own projections. In this narrative, even though Rudra himself is a projected subject, he is not only projected to be self-realized, but also causes realization for the other subjects. The case of Lavaṇa's or Gādhi's subjectivities

demonstrates a shift back to an early or more fundamental subjectivity rather than a total awakening. In all contexts, what is changing is the very modes of subjectivity. But is this subjectivity actually shifting? From the perspective of the liberated one, nothing is ever changing, but from the relative or phenomenal perspective, difference is fundamental and change is real. As long as there is the dyad of subject and object, there is a shift in subjective modes, but from the perspective of the realized one, there is no shift. The first two narratives, as we can see, epitomize parallel subjectivities, as they are not simply negating the manifest modes of subjectivity but rather confirming parallel lines of subjectivity with different temporal streams. The following stanzas from the narrative of one hundred Rudras confirm the same conceptual parameters:

Just like tides are the motion of water, the world is the same with regard to consciousness. [For] Rāma, the only difference here is in the case of tides in water, [being] possible only upon the being of time, space, and structure, whereas in the case of the world, space, etc. [these] are seen even though they do not exist[12].

Accordingly,

All exists within the treasury of consciousness. In whatsoever way consciousness observes something, it actualizes that accordingly, since the one is composed of plurality[13].

For YV, being one or many, enduring change or remaining as changeless, are simply the varied modes of a singular consciousness, like facets of a single gemstone. Accordingly, the paradoxicality of being many and one, changing while remaining singular, is what constitutes the intrinsic nature (*svabhāva*) of the absolute:

Since the embodied subject is of the character of having a singular power, for [the one is] intrinsically of the character of all the powers. On its own, its nature is of endless [diversity] while at the same time, non-differentiated [singularity][14].

If we read the narratives of Lavaṇa or Gādhi, we are repeatedly reminded of the same underlying assumption of the singularity that always encompasses plurality, or of the plurality that contains the identical underlying principle within. Just like in the "relativity" of M. C. Escher, the world of YV exists outside of the governing physical laws that we consider as constant. YV repeatedly reminds us that all that exists is just one, but at the same time, it also confirms that this absolute is dynamic and the manifoldness is its inherent nature. What makes the world of YV interesting is the surreal nature of what is given, making paradoxicality corresponding to the one and the many as the most rational conclusion. Rather than the concept of *māyā* being used to negate what is phenomenal, it is used here to explain the indeterminacy of the manifold that does not violate singularity while projecting diversity. In essence, the power of *māyā* that is intrinsic to Brahman makes this paradoxicality an everyday reality. For instance:

It is just due to appearance that everything revolves around. Even a single moment turns into an eon and an eon turns into a moment[15].

In the narrative of Gādhi, YV stresses accidentalism, where all that appears has no specific reason and everything becomes possible due to the indeterminacy of *māyā*. This coincidence is utilized to explain intersubjectivity: a single mirage can be perceived by many, a single play can be enjoyed by many children, many deer can be confused by the green that appears like grass in the forest, and that, even for many, a single intuitive experience can arise[16]. Thus, YV gives the philosophy of temporality on this ground that:

Brahmin, what you have heard —-time is what makes suspension and permission possible—is nothing but pure conceptualization. Time as such resides within the self. The all-powerful (*bhagavān*) time is formless, birthless, and is identical to the Brahman. It neither negates anything from anybody ever, nor does it affirm anything. On the other hand, phenomenal time in the form of year or the eons of *kalpa* or *yuga*, is conceived of by the collection of entities and it makes it possible for the synthesis of entities[17].

The above passage endorses three central arguments. First, it rejects the position of Bhartṛhari that the function of time is to suspend and permit the manifestation of specific tropes, on the basis of which concepts such as growth and decay are possible[18]. Second, it does not negate time as such but it identifies time with the self or consciousness. And third, it grounds the argument that phenomenal time is relational, and that time only explains the interrelationship among entities. Accordingly, phenomenal time exists only in relation to the corresponding entities, and it is due to this measuring device of time that entities are temporally determined. In essence, there is no actual permission or suspension—a real causal flow—and the causality that is imposed on the basis of growth and decay is merely relational and does not have any absolute reality. Finally, absolute time is nothing other than the very self of the character of consciousness.

## 6. Revisiting Gaudapāda in Light of *Yogavāsiṣṭha* Narratives

Gauḍapāda has been criticized as crypto-Buddhist for his appropriation of Mādhyamika and Yogācāra arguments, and any uncritical gaze leads a reader to the same conclusion[19]. There is no argument that Gauḍapāda adopts the logical framework of the Mahāyāna Buddhists and also utilizes many of their examples. However, we would be gravely mistaken if we failed to recognize the philosophical parameters that Gauḍapāda is not willing to breach in accepting the logical framework from his Buddhist counterparts. And for this, I would like to focus on a single term, *svabhāva*, meaning intrinsic nature, or the mode of being what it is. Gauḍapāda will not breach the parameters of *svabhāva*, where in his case, consciousness, as such, the Brahman, has a self-presenting nature and is singular. The foundational argument of Nāgārjuna rests on the very negation of *svabhāva*. When Gauḍapāda rejects change, he is not rejecting the changeless basis having its own intrinsic nature. However, if Nāgārjuna were to also affirm this changeless basis, his philosophy would be endorsing *svabhāva*. This would not only contradict the thesis of "no inherent nature" (*niḥsvabhāva*) but would also collapse his rejection of the Sarvāstivāda doctrine. When Nāgārjuna rejects time or change, he is not endorsing something timeless and changeless. Instead, what he is arguing is that the very notion of temporality or change lacks its own intrinsic nature. His is the argument that there exists nothing that is timeless or changeless on its own. He argues that temporality or change dependently arise (*pratītya-samutpāda*) and for this reason, lack their own "being-what-it-is-ness" (*svabhāva*). On the other hand, when Gauḍapāda rejects temporality, he is affirming the non-temporal and changeless substratum, the absolute, the Brahman, the unmodified consciousness which he accepts as the basis for all that manifests. While they both argue for the absolute reality (*paramārtha*), what they mean by it is not identical. For Gauḍapāda, *paramārtha* refers to something absolutely real that is covered by the veil of *māyā*, and once the curtain is removed, one recognizes oneself as this basis of the character of luminous consciousness. For Nāgārjuna, on the contrary, the realization of *paramārtha* refers to the recognition of the co-constitution of all that can be posited. If Nāgārjuna were to confirm a positive substratum, he would only be advocating for the Brahman in some other terms such as *śūnyatā*. But his is the claim that reality as such is characterized by an emptiness or a lack of intrinsic nature (*niḥ-svabhāva*)[20]. On the other hand, if Gauḍapāda were to read negation of the phenomenal in terms of pure negation, he would be collapsing Advaita with Mādhyamika. What Gauḍapāda is negating is the projection or the motion picture visible upon the screen to confirm the screen upon which lies projected or phenomenal reality. The movement seen in the motion picture is not the movement of the screen, Gauḍapāda would argue. Nāgārjuna would argue that the concepts of the screen and the motion picture are interrelated and lack their own original nature.

As the focus of this paper is Gauḍapāda, I will briefly address only the relevant categories from the central work of Nāgārjuna, the *Mūlamādhyamikakārikā* (MMK). Three small sections in this text provide sufficient information to ground Nāgarjuna in this context: the section on change (Section 2)[21], the section on *svabhāva* (Section 15) and the section on time (Section 19). The essence of what Nāgārjuna says in this first section is that moving as

such cannot be independently verified and motion is dependent upon something moving. To say that X is moving, we need to determine X and the paradoxes that follow: whether we determine X as non-moving wherein motion is imposed, or whether X itself is not determined as it is changing, and movement or change is imposed upon something that is yet to be determined. This means that, in the instant we determine something, change is overlooked, and when we address change, the determined entity is not there. In the section on *svabhāva* (Section 15), Nāgārjuna extends his arguments by saying that *svabhāva* as such cannot be grounded by some cause that is external to itself, for that would not amount to the inherent nature of what it is. All that manifests being relative to the other lacks its own inherent nature and is therefore devoid of self-nature. However, if an entity exists due to the existing nature of factors external to itself, that would be assuming the external nature in the absence of self-nature, and even that would be contradictory for the reason that, in that case, what amounts to being external would be required to have its own *svabhāva*[22]. His rejection of *svabhāva* is the rejection of *bhāva* or being as such, and this is not an affirmation of absence (*abhāva*), for he extends the same argument that if there is nothing to posit to begin with, there is nothing to negate either. Finally, in the section on the examination of time, Nāgārjuna explores the modes of temporality and demonstrates that they lack their inherent nature. The essence of these arguments is that none of the modes of past, present, or future are self-given, and the very notion of temporality itself depends on circularity. In essence, the fundamental thrust behind Nāgārjuna is to counter the Sarvāstivāda arguments[23], and it so happens that there is an overlap between some of the Sarvāstivāda categories and Hindu philosophical schools. The concept of *svabhāva* broadly overlaps these schools, even though what is meant by *svabhāva* differs as we examine each of the schools, be it Sāṅkhya, Advaita, or Trika-Pratyabhijñā.

This is to say that we have to read Gauḍapādiyan analogies and arguments again, keeping in mind the central categories such as the intrinsic nature (*svabhāva*), and not reduce philosophies by finding similarities in their vocabulary or style of argumentation. I have introduced narratives from YV in this conversation for the additional reason that YV also repeatedly endorses *svabhāva* or intrinsic nature. On the one hand, YV is rejecting the phenomenal by using dream analogies. On the other hand, YV confirms that dreams and fantasies are real in some planes of existence. In the above narratives, all of the protagonists confirm the real existence of what they thought was a mere fantasy, breaching the gap between the virtual and the actual. When Gauḍapāda compares waking reality with dreaming, we are reading these instances as merely suggesting the negation of the phenomenality of everyday consciousness. It is noteworthy that YV uses dream analogies not just to negate the phenomenality of everyday consciousness, but also to affirm the being of dream reality, that all events in the virtual world are real events, and that they do indeed bear the same degree of reality as do events in the actual world. There is no reason why we cannot read Gauḍapāda along the same lines, that the phenomenal and the dream bear the same degree of reality and that this reality is subsumed within the foundational reality of the self or Brahman. But, even then, there is a shift in focus. While both explain reality in two degrees by collapsing the difference between dreams and waking states, if YV is using dream analogy to explain differentiation, Gauḍapāda is instead using the same to negate the substantiality of waking reality. Moreover, YV uses the same analogy to ground the fantastic as bearing the same degree of reality as the phenomenal[24], meaning that rather than negating the substantiality of the waking reality, the dream analogy becomes a means to affirm the substantiality of dreaming. We come to the conclusion when reading verses like the following that Gauḍapāda is using negation to affirm the absolute:

> If the manifold were to exist, without a doubt, it would also disappear. This duality is nothing but *māyā*. In the sense of absolute reality, it is non-dual...[25]

It is evident that this is not what Nāgārjuna has maintained. We therefore need to keep in mind the points of divergence when we read Gauḍapāda in light of earlier Buddhist philosophers such as Nāgārjuna and Vasubandhu. In all accounts, for Nāgārjuna, negation is negation and not an affirmation. If *svabhāva* arguments are central to YV in affirming

the manifold without violating foundational non-duality, Gauḍapāda is arguing for the same. An additional reason for situating *svabhāva* in this conversation is that because when Nāgārjuna claims "emptiness" (*śūnyatā*), this is not a non-qualified term but rather "the emptiness of having no intrinsic nature" (*niḥ-svabhāva*). Even though the treatise of Gauḍapāda is relatively a small text, it gives ample references. Before reading the examples, also keep in mind that these examples are predominantly borrowed from the fourth chapter:

- "The [manifestation of the manifold] is the *svabhāva* of the luminous being"[26].
- "Duality is merely *māyā*. It is nonduality in the absolute sense (*paramārthataḥ*)"[27].
- "The absolute reality (*paramārthatā*) is that there is neither a seeker nor a liberated being"[28].
- "Non-duality is the absolute reality whereas duality is its [self-]differentiation'[29].
- "There never is turning into the other (*anyathābhāva*) with regard to the essential nature (*prakṛti*)"[30].
- "Those who argue that the deathless turns into being mortal by its own nature (*svabhāva*), since the absolute will be originated, how will it remain changeless?"[31]
- "Essential nature (*prakṛti*) should be recognized as never rejecting its essential being (*svabhāva*) as self-accomplished, having its own intrinsic mode of being (*svābhāvikī*), inborn (*sahajā*), and non-constructed"[32].
- "All the dharmas are free from old age and death [because of] their own essential mode of being (*svabhāva*)"[33].
- "The cause does not come into being from some beginningless result. Neither does the result come into being on its own (*svabhāvataḥ*)"[34].
- "There never is contradiction within the essential nature (*prakṛti*)"[35].
- "That what exists in the limited mode of conceptualization' (*kalpita saṃvṛti*) does not exist in the sense of the absolute (*paramārthataḥ*)"[36].
- "While even to say 'birthless,' [Brahman] is relying on the mode of limited conceptualization, in the absolute sense, it is not even birthless"[37].
- "The self is constantly luminous due to its own nature"[38].
- "It is self-given that all individual selves are by their own nature (*prakṛti*) inherently self-aware"[39].
- "All the selves are devoid of covering and are stainless in their intrinsic nature"[40].

This repeated application of the vocabulary of *svabhāva*, *prakṛtri*, or *paramārtha*, all positively affirming something foundational as having its own intrinsic nature, helps us confirm that for Gauḍapāda, there exists something as the substratum of projected reality that is changeless, and this very changelessness constitutes the core nature of what it is. One can argue at this juncture that Nāgārjuna has already established a two-tier structure of explaining reality and Gauḍapāda is simply adopting this. This is true. But this does not change the fact that for Nāgārjuna, there exists nothing having its own intrinsic nature, something as a changeless basis, the substratum for the projected reality. On the other hand, Gauḍapāda is confirming the opposite, that there only exists the absolute, the Brahman, and the manifold projected on it lacks originality. For Nāgārjuna, there is not a thing posited to be unchanging, whereas for Gauḍapāda, this is the case. Something positive should not be extracted as the conclusive meaning from Nāgārjuna's negations. However, when Gauḍapāda negates, he is establishing the positive substratum, the Brahman, that is outside of the scope of negation. When Nāgārjuna negates both subject and object, he is pointing out their interdependent nature. However, when Gauḍapāda negates the same, he is affirming the basis upon which these two polarities are projected. For Nāgārjuna, the conceptualized (*kalpita*) is co-constituted, meaning both subject and object are constituting themselves being dependent on each other. However, for Gauḍapāda, the foundational changeless consciousness becomes the subject of experiencing its own projected reality. In essence, even though Gauḍapāda does appropriate several categories from Nāgārjuna and other Mahāyāna Buddhists, he is uncompromisingly standing on the ground of the non-dual Brahman, the non-temporal consciousness, upon which manifests duality. Both may be using the term *advaya* or non-dual, but by this term, Buddhist

philosophers such as Nāgārjuna or Asaṅga are rejecting duality in terms of the subject and object, whereas Gauḍapāda is confirming the singularity of Brahman, the self-given consciousness[41]. Even though they both confirm that change as such does not have an absolute nature, if for Nāgārjuna there is nothing to change, for Gauḍapāda, what exists as the basis does not change.

So far, I have only addressed the rejection of *svabhāva* in the Madhyamika of Nāgārjuna. A valid objection to this reading can be raised that the Yogācāra school does accept *svabhāva* and Gauḍapāda is anyway heavily borrowing categories from the Yogācāra—so why not credit the concept of *svabhāva* to the Yogācāra, particularly as argued by Asaṅga and Vasubandhu? This argument has some validity because there is indeed a text, *Trisvabhāvanirdeśa* (TSN), attributed to Vasubandhu, and the concept of *svabhāva* is ubiquitous if we reach out to the Sarvāstivāda school and the Buddhist sub-schools under the influence of Pudgalavāda. Concepts such as *tathāgatagarbha* or *dharmadhātu* or even *amalavijñāna* of Paramārtha convey an unmistakable mark of the centrality of the being of something, or affirmation of the basis, and this comes very close to the Gauḍapāda's concept of *svabhāva*. As far as the *svabhāva* in the Yogācāra and, in particular, the concept of *pariniṣpanna svabhāva* in the TSN is concerned, the *svabhāva* confirmed here is negation; it does not affirm the being of something, but rather is used in negation. The consummate nature (*pariniṣpanna svabhāva*) is here defined as: "The constant state-of-not-being-found of "how it appears" in that which appears, can be known as the fulfilled own-being, because of its state of non-otherness"(Anacker 2005, p. 291). TSN proclaims, "*sattvena gṛhayte yasmād atyantābhāva eva ca |*" (verse 11ab), or "the *kalpita* nature is what does not exist in the absolute sense but nonetheless [is] grasped as existing". Accordingly, "*vidyate bhrāntibhāvena yathākhyānaṃ na vidyate |*" (TSN 12ab), or "the *paratantra* is what is grasped due to error and has no name to it". And, likewise, "*advayatvena yac cāsti dvayasyābhāva eva ca*" (TSN 13ab) or the "perfectly attained form" (*pariniṣpanna svabhāva*) is "what exists in the mode of the non-dual as the absence of the dual". In essence, it is negation that constitutes the core of *svabhāva* in the TSN, and not in the Gauḍapādian sense of an affirmation of being. What we can say conclusively regarding the three *svabhāvas* in TSN is summed up in a single verse:

> *kalpito vyavahārātmā vyavahārātmako paraḥ |*
>
> *vyavahārasamucchedaḥ svabhāvaś cānya iṣyate | |* TSN 23.
>
> That which constitutes the conventional is of the character of conceptualization. The other one [or the interdependent *svabhāva* is] also of the character of conventionality. The other [or the perfectly attained form] is of the elimination of the conventional.

We can thus conclusively say that the *svabhāva* that we encounter in the Yogācāra literature after Vasubandhu is of the character of negation, and is broadly compatible with the Mādhyamika metaphysics of emptiness. As we know from the above conversation, this is the opposite of what Gauḍapāda has in mind when it comes to his arguments regarding *svabhāva*. It is therefore advisable to read the triadic structure of the phenomenal presentation in terms of *lakṣaṇa*, or characteristics rather than *svabhāva* or 'intrinsic nature,' and this is the case that the early reference for the concept as we can glean from the *Sandhinirmocanasūtra* comes as *tri-lakṣaṇa* rather than *tri-svabhāva* (Ming-Wood 1982).

Returning to the narrative philosophy of the *Yogavāsiṣṭha*, both Gauḍapāda and the author of the YV accept that there exists the basis for duality. However, if for Gauḍapāda this basis is absolutely changeless and duality is projected due to illusion, YV considers the manifold as an inherent nature of the absolute. The main difference is, YV focuses on the manifest modes of the absolute in order to explain the *svabhāva* of the absolute in expressing into the manifold without violating its primordial singularity, whereas Gauḍapāda is using the analogy of dreams or illusions merely to negate the substantiality of what is projected. For YV, it is the *svabhāva* of consciousness and, as such, to be revolving that means that manifoldness and singularity are both possible at the same time, whereas for Gauḍapāda, dichotomies are projected, unreal, and the essential nature is bereft of the potencies to

express itself into the manifold. Even though both Gauḍapāda and the author of YV endorse the fundamental nature or *svabhāva*, what they each mean by *svabhāva* is fundamentally different. For YV, paradoxicality in embracing singularity while also confirming change is not a problem, as this very paradoxicality is what *māyā* represents. On the other hand, since duality is merely a projection of *māyā* and in reality there is no projection, there is no paradoxicality in the paradigm of Gauḍapāda.

In essence, when Nāgārjuna negates temporality or change, he is pointing out that time or change as such does not have its own inherent nature, as the sense of temporality or the sense of change arises being dependent upon something temporal or something in flux. When Gauḍapāda negates time and change, he is establishing the non-temporal and changeless nature of the absolute, the Brahman, of the character of pure consciousness. And when YV negates temporality or change, it is only suggesting that manifestation of the manifold does not violate the singularity of the Brahman of the character of pure consciousness. Gauḍapāda is not endorsing paradoxicality, as there is no paradoxicality in the singular absolute. However, YV is endorsing paradoxicality by maintaining that the singularity of the absolute is not violated by the manifest modes of the manifold.

**Funding:** This research received no external funding.

**Data Availability Statement:** No new data were created or analyzed in this study. Data sharing is not applicable to this article.

**Acknowledgments:** This research was made possible by SN Sridhar and Kamal Sridhar by providing accommodation during the research at Stony Brook University.

**Conflicts of Interest:** The author declares no conflict of interest.

## Notes

[1] I am using the term "dialogue" in a loose sense, engaging some of the categories of Nāgārjuna and Gauḍapāda. While Nāgārjuna offers a brief section on temporality, Gauḍapāda does not even do that, making it very difficult to ground the philosophy of time by bringing these two philosophers into a virtual dialogue (*saṃvāda*). Nevertheless, these two philosophers have shaped two major philosophies in India: Mādhyamika Buddhism and Advaita Vedanta.

[2] BL Atreya pioneered this line of arguments (see Atreya 1936, 1993).

[3] For the status of imagination in Gauḍapāda's philosophy, see Timalsina (2013). For the positive role of imagination in Indian classical philosophy, see (Timalsina 2020).

[4] Sanskritist linguist philosophers have identified two types of negation: *prasajya-pratiṣedha*, or direct negation, and *paryudāsa pratiṣedha*, the negation of something in order to affirm something positive. For addressing this issue, see Timalsina (2014a). In reading Nāgārjuna, the Prāsaṅgika and Svātantrika approaches broadly rely on the aforementioned difference in interpreting negation. For the two-truth doctrine in Mādhyamika philosophy, read Eckel (1992). For the current discourse in interpreting Nāgārjuna, read Ferraro (2013a, 2013b), and Siderits and Garfield (2013). For the Prāsaṅgika reading of Nāgārjuna, read Garfield (2006).

[5] *Vigrahavyāvartanī*, verse 29. See Johnston and Kunst (1978).

[6] There is no denying that Gauḍapāda uses the argumentative style, some of the arguments, and some vocabulary from the Mahāyāna literature. See Wood (1990), Bhattacharya (1992), or King (1995) for this. A negative consequence of this trend of reading has emerged to overlook the originality of Gauḍapāda's thought. A thorough study is needed to analyze the ways Gauḍapāda appropriates the Buddhist Mādhyamika and Yogācāra arguments to buttress his philosophical claims.

[7] *jāyamāno hi ced dharmo gṛhyate kathaṃ tasmāt pūrvaṃ kāraṇaṃ na gṛhyate? avaśyaṃ hi jāyamānasya grahītrā tajjakaṃ grahītavyam* | Śaṅkara upon GK IV.21.

[8] *ātmā hy ākāśavaj jīvair ghaṭākāśair ivoditaḥ* | *ghaṭādivac ca saṅghātair jātāv etan nidarśanam* | | GK III.3.

[9] *yathā svapne dvayābhāsam spandate māyayā manaḥ* | *tathā jāgrad dvayābhāsam spandate māyayā manaḥ* | | GK III.29.

[10] We need to keep in mind that YV does not make a distinction between actual and virtual events, and so the words for narrative, *ākhyāna* or *ākhyāyikā*, or *kathā*, appear interchangeable with history or *itihāsa* (YV, Nirvāṇa I.62.1).

[11] For an extensive treatment of this concept, see Timalsina (2006).

[12] *vīcir yathāmbhasaḥ spando jagac caiva tathā citau* | *etāvanmātra evātra bhedo yad raghunandana* | | *deśakālasvarūpeṣu satsu vīcyāditāmbhasi* | *jagadādau tu deśādyā asanto jagatīkṣitāḥ* | | YV, Nirvāṇa I, chapters 72–73.

[13] sarvam asti citaḥ kośe yad yathālokayaty asau | cit tathā tad avāpnoti sarvātmatvād avikṣatam | | YV, Nirvāṇa I. chapter 64, verse 13cd–14ab.

[14] sarvaśaktyaḥ svarūpatvāj jīvasyāsty ekaśaktitā | anantaś cāpṛthaktaś ca svabhāvo 'sya svabhāvataḥ | | YV, Nirvāṇa I 64.26.

[15] pratibhāsavaśād eva sarvo viparivartate | kṣaṇaḥ kalpatvam āyāti kalpaś ca bhavati kṣaṇaḥ | | YV, Utpatti 121.18.

[16] *kadācit pratibhaikaiva bahūnām api jāyate* | YV, Upaśama 49.10ab.

[17] pratibandhābhyanujñānāṃ kālo dāteti yā śrutiḥ | vipra saṃkalpamātro 'sau kālo hy ātmani tiṣṭhati | | YV, Upaśama 49.14.amūrto bhagavān kālo brahmaiva tam ajaṃ viduḥ | na jahāti na cādatte kiñcit kasya kadeti ca | | YV, Upaśama 49.15.laukiko yas tv ayaṃ kālo varṣakalpayugātmakaḥ | saṃkalpyate padārhaughaiḥ padārthaughaś ca tena tu | | YV, Upaśama 49.16.

[18] For this, see the Kālasamuddeśa (*Vākyapadīya*), verse 4.

[19] Vidhushekhar Bhattacharya (1992) championed this line of arguments in his critical study and commentary upon the *Gauḍapāda-kārikā*. S. N. Dasgupta has entirely adopted this position in his voluminous work. See Dasgupta (1922, vol I, pp. 423–29). Karmarkar (1953) has briefly responded to these objections.

[20] I am here referring to section 15 of MMK, where Nāgārjuna categorically rejects *svabhāva*.

[21] Kalupahana (1991) translates the chapter as "examination of the moved and the not-moved" for the section: *gatāgataparīkṣā*. The fact of the matter is, this rejection of something mobile and the absolute as immobile counters the *svabhāva* thesis that the absolute is changeless. And so, this section is better understood if we read it as addressing change and changelessness.

[22] For a brief introduction on Nāgārjuna's critique of *svabhāva* and for references on the same topic, see Westerhoff (2022).

[23] I am in agreement with Kalupahana (1991) in forwarding this argument.

[24] For manifestation or projection (*ābhāsa*) in the *Yogavāsiṣṭha*, see Timalsina (2014b).

[25] prapañco yadi vidyeta nivarteta na saṃśayaḥ | māyāmātram idaṃ dvaitam advaitaṃ paramārthataḥ | | Māṇḍūkya Kārikā I.17.

[26] *devasyaiṣa svabhāvo 'yam ...* | GK I.9a.

[27] *māyāmātram idaṃ dvaitam advaitaṃ paramārthataḥ* | GK 1.17cd.

[28] *na mumukṣur na vai mukta ity eṣā paramārthatā* | GK 2.32cd

[29] *advaitaṃ paramārtho hi dvaitaṃ tadbheda ucyate* | GK 3.18ab.

[30] *prakṛter anyathābhāvo na kathañcid bhaviṣyati* | GK 4.7cd.

[31] svabhāvenāmṛto yasya dharmo gacchati martyatām | kṛtakenāmṛtas tasya kathaṃ sthāsyati niścalaḥ | | GK 4.8.We need a separate critical analysis for the application of dharma in Advaita literature. Gauḍapāda does not appear to be using this term as possessing a single meaning. It seems that he uses it in some contexts in the same way Mahāyāna texts utilize dharma, in others, as a generic term for all the properties, and on some occasions, to refer to the self. This last meaning is derived based on the way Śaṅkara's commentary reads the passages under consideration.

[32] sāṃsiddhikī svābhāvikī sahajā akṛtā ca yā | prakṛtiḥ seti vijñeyā svabhāvaṃ na jahāti yā | | GK 4.9.

[33] *jarāmaraṇanirmuktāḥ sarve dharmāḥ svabhāvataḥ* | GK 4.10ab.

[34] *hetur na jāyate 'nādeḥ phalaṃ cāpi svabhāvataḥ* | GK 4.23ab.

[35] *prakṛter anyathābhāvo na kathañcid bhaviṣyati* | GK 4.29cd.

[36] *yo 'sti kalpitasaṃvṛtyā paramārthena nāsty asau* | GK 4.73ab.

[37] *ajaḥ kalpitasaṃvṛtyā paramārthena nāpy ajaḥ* | GK 4.74ab.

[38] *sakṛdvibhāto hy evaiṣa dharmo dhātusvbhāvataḥ* | GK 4.81cd.

[39] *ādibuddhāḥ prakṛtyaiva sarve dharmāḥ suniścitāḥ* | GK 4.92ab. Also, GK 4.93 continues the same conversation on *prakṛti*.

[40] *alabdhāvaraṇāḥ sarve dharmā prakṛtinirmalāḥ* | GK 4.98ab.

[41] The only place the exact term *advaya* appears in Nāgārjuna's work is in *Bodhicittavivaraṇa* (McCagney 1997, p. 128); however, this term appears more frequently in the Yogācāra.

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
