# Peer review of "Time and Change in Advaita—Gauḍapāda in Dialogue with Vasiṣṭha and Nāgārjuna"

_religions, doi:10.3390/rel15020167_

Round 1
Reviewer 1 Report
Comments and Suggestions for Authors
This is an interesting paper addressing the important theme of time and change in Advaita. It is well-written and almost ready to be published. I have three points for the author to consider:
1) A main point on the Buddhist influence on Gauḍapāda, which is one of the main concerns of the current paper. The author has argued convincingly that Gauḍapāda is different from Nāgārjuna with regard to the key concept of svabhāva. This may distance him from Madhyamaka, but he still could be a crypto-Buddhist for his appropriation of Yogācāra philosophy. To my understanding, Gauḍapāda comes close to the Yogācāra position in many respects: positive usage of svabhāva (as in their foundational theory of trisvabhāva), affirming the absolute status of consciousness, singling out the dualistic structure of subject-object when negating the illusory phenomena. The author may tune down a bit by holding that Gauḍapāda can be distinguished from Madhyamaka, but share views with Yogācāra. In this sense, he is still a crypto-Buddhist.
2) The phrase “svabhāva-śūnyatā” is used twice in the paper. To my knowledge, this expression is not found in Buddhist writings. Is it attested in Gauḍapāda or other Hindu author? If not, please use “niḥsvabhāva” instead.
3) Page 7 line 325 seems to be the beginning of a new section. This may be a mistake of typesetting.
Author Response
I have carefully revised the document keeping in mind all your comments. Attached to this email is the revised paper.

Reviewer 2 Report
Comments and Suggestions for Authors
No major issues, except for two instances where sources should be referred to explicitly.
See two notes on the PdF.

Author Response

(The authors gave the same response as above.)

Reviewer 3 Report
Comments and Suggestions for Authors
Author Response

(The authors gave the same response as above.)

Reviewer 4 Report
Comments and Suggestions for Authors
1. The content of the essay is appropriate for this special issue, and it is certainly an answer to this topic from the viewpoint of advaita, but it is a rather casual essay. In other words, one of the features of this essay is to compare advaita with Nāgārjuna, and there are frequent references to him. However, there is only one specific reference to his own work: Vigrahavyāvartanī. However, from the content of the paper, it is likely that the author has MMK in mind as Nāgārjuna's thought, although he does not mention it in any of this draft. The author should present his own interpretation of MMK by citing specific MMK phrases when he refers to Nāgārjuna.
Reading Nāgārjuna along these lines would violate all the fundamental principles of 522
critical analysis.
☞There will be readers who read Gauḍapāda in the context of Nāgārjuna, but not readers who read Nāgārjuna in the context of Gauḍapāda.
Both may be using the term advaya or non-dual, but by this term, 577
Nāgārjuna is rejecting duality in terms of subject and object whereas Gauḍapāda is con- 578
☞Please list specific texts and phrases in which Nāgārjuna uses the word advaya.
But his is the claim that reality as such is characterized by the emptiness or the 488
lack of intrinsic nature (svabhāva-śūnyatā).
☞Please list specific texts and phrases in which Nāgārjuna uses the word svabhāva-śūnyatā.
2. Please indicate at the beginning that MUK and GK are identical.
3. fn. 2 Sanskritist linguist philosophers have identified two types of negation: prasajya, or direct negation, and pratiṣedha
☞The two types of negation are prasajyapratiṣedha and paryudāsa.
4. There are many typos in Sanskrit.
fn6 hyākashavaj>hy ākāśavaj
fn7 jāgraddvayābhāsam> jāgrad dvayābhāsam
fn9 citau>citau(italisize u)
fn10 sarvamasti>sarvam asti, tatha>tathā
fn 11 jīvasyāstyekaśaktitā>jīvasyāsty ekaśaktitā
fn14 kāḷo>kālo
fn21 ityeṣā>ity eṣā
Line536 fn23 anyathābhāvo means change, not contradiction
fn25 jahātI>jahāti
fn26 dharmaḥ>dharmāḥ
fn29 kalpitasaṃvṛttyā>kalpitasaṃvṛtyā, nasty>nāsty
fn31 dhātusvāvataḥ>dhātusvabhāvataḥ
fn33 dharma>dharmāḥ
☞In fn26, the word dharmāḥ is translated as dharmas, but in fn 31, 32, 33, it is translated as self or selves. What is the rationale for the different translations? Or should they be unified?
5. Other minor typographical errors are often found. I just mention some of them.
Journal of Indian Philosophies>Journal of Indian Philosophy
maya>māyā
abhuta>abhūta
Line58 Māṇḍūkya 58 Upaniṣad (MUK)>Māṇḍūkya 58 Upaniṣad (MU)
Gauḍapada>-pāda
Māṇḍukya>-ūkya
Author Response

(The authors gave the same response as above.)

Round 2
Reviewer 4 Report
Comments and Suggestions for Authors
This is an interesting paper and most of the minor points I pointed out have been improved. Only a few more typographical errors are noted.
fn4 two types of negation: prasajya-pratiṣedha, or direct negation, and pratiṣedha
☞again, the second should be paryudāsa.
please specify a or b of Timalsina 2014.
fn8 ākashavaj
☞again, ākāśavaj
fn19 karikā>kārikā
the emptiness 581
of having any intrinsic nature’
>having no intrinsic nature
reference8 losophies>losophy
Sometimes different fonts are mixed.
Author Response
Thank you so much for reading the paper very closely. Attached, the revised paper.
